# ENHANCING GRAPH NEURAL NETWORKS: A MUTUAL LEARNING APPROACH

## ABSTRACT

Knowledge distillation (KD) techniques have emerged as a powerful tool for transferring expertise from complex teacher models to lightweight student models, particularly beneficial for deploying high-performance models in resource-constrained devices. This approach has been successfully applied to graph neural networks (GNNs), harnessing their expressive capabilities to generate node embeddings that capture structural and feature-related information. In this study, we depart from the conventional KD approach by exploring the potential of collaborative learning among GNNs. In the absence of a pre-trained teacher model, we show that relatively simple and shallow GNN architectures can synergetically learn efficient models capable of performing better during inference, particularly in tackling multiple tasks. We propose a collaborative learning framework where ensembles of student GNNs mutually teach each other throughout the training process. We introduce an adaptive logit weighting unit to facilitate efficient knowledge exchange among models and an entropy enhancement technique to improve mutual learning. These components dynamically empower the models to adapt their learning strategies during training, optimizing their performance for downstream tasks. Extensive experiments conducted on three datasets each for node and graph classification demonstrate the effectiveness of our approach.

## 1 INTRODUCTION

Graph Neural Networks (GNNs) have emerged as powerful tools for learning representations of structured data and for the extraction of node and graph embeddings to facilitate a wide range of graph mining tasks, including node classification, link prediction, and graph classification Kipf and Welling (2016); Hamilton et al. (2017); Borisyuk et al. (2024); Schlichtkrull et al. (2018). GNNs generally leverage the message-passing framework, where nodes aggregate information from their neighbors to capture information about node features and the underlying graph structures.

In deep learning (DL), knowledge distillation (KD) methods have been instrumental in balancing model size and accuracy. These methods involve transferring knowledge from complex teacher models to smaller student models, enabling the student model to emulate the pre-trained teacher's logits and/or feature representation, thereby matching or surpassing the teacher's performance. However, applying KD to GNNs presents unique challenges, particularly due to their typically shallow architectures and over-smoothing issues Li et al. (2018).

In this study, we demonstrate a departure from the traditional approach of distilling knowledge from a teacher GNN to a student GNN and instead propose a mutual learning approach to train small but powerful GNNs. Our motivation partly comes from a recent study Guo et al. (2023), which demonstrates that GNNs can encode complementary knowledge owing to their diverse aggregation schemes. Furthermore, mutual learning involves a collective training process where untrained student models collaboratively work to solve a task Zhang et al. (2018). This collaboration entails matching alternative likely classes predicted by other participants to increase each participant's posterior entropy, ensuring better generalization during testing Pereyra et al. (2017). The rationale behind mutual learning lies in the fact that each model starts training from a different initialization and is guided by its supervision loss. This individualized guidance and initialization ensures that the models avoid learning identical representations, even when predicting the same labels.

Despite the suitability of graph learning techniques for numerous large-scale industrial applications, MLPs remain prevalent for various prediction tasks within this domain Zhang et al. (2021). Zhang et al. (2021) demonstrate that MLPs can effectively learn from pre-trained GNNs, suggesting that the disparity in expressive power between GNNs and MLPs is often negligible in real-world scenarios. Therefore, we leverage mutual learning to improve performance across GNNs and subsequently showcase that this knowledge is transferable to MLPs that are suitable for latency-constrained industrial applications.

In this work, we (1) investigate the feasibility of cooperatively training multiple GNNs, (2) propose enhancements in collaboration to ensure that each participant prioritizes crucial knowledge, an aspect often overlooked in conventional deep mutual learning, and (3) explore the transferability of representations acquired during collaboration by each target model for KD. To achieve these goals, we introduce a novel framework, Graph Mutual Learning (GML), designed to collectively train a set of untrained GNNs. Our framework promotes and enables collaborative learning and knowledge sharing among peers, resulting in improved performance compared to isolated training. To enhance generalization, we incorporate the confidence penalty mechanism Pereyra et al. (2017) to penalize low-entropy output distributions. We also propose an adaptive logit weighting scheme to allow each model to focus on essential knowledge during the mutual learning process for efficient learning. Finally, beyond mutual learning, we adapt GML for KD, ensuring that the representation acquired during collaborative training can be easily transferred to a student MLP for faster inference.

To summarize, our contributions are: (1) We employ mutual learning to collectively train a group of GNN peers. This approach enhances the performance of individual models by promoting collaborative learning and knowledge sharing. (2) We introduce an adaptive logit weighting scheme to efficiently prioritize crucial knowledge during collaborative training, enhancing the efficiency of the learning process. (3) We adapt the GML framework for knowledge distillation, ensuring that the representations acquired during collaboration are versatile and readily transferable to other models.[1] (4) We evaluate the effectiveness and performance of our approach using publicly available datasets for node and graph classification tasks. Empirical results show that GML significantly improves the performance of shallow GNNs for different tasks.

## 2 RELATED WORK

**Graph Neural Networks.** Early research in GNNs laid the foundation for their application in various domains, including social networks, bioinformatics, and recommendation systems Borisyuk et al. (2024); Kang et al. (2022); He et al. (2020). Techniques such as recursive neural networks applied to directed acyclic graphs Frasconi et al. (1998); Sperduti and Starita (1997) paved the way for the development of GNNs. Generally, GNNs employ a message-passing framework, utilizing an iterative approach to aggregate neighborhood information. This process entails nodes aggregating feature vectors from their neighbors to compute their updated feature vectors Xu et al. (2018a;b); Gasteiger et al. (2018). Kipf and Welling (2016) made key contributions by introducing the Graph Convolutional Network (GCN), a novel architecture tailored for graph data. GCNs estimate node embeddings by aggregating information from neighboring nodes and applying a self-loop update technique. Similarly, Hamilton et al. (2017) introduced GraphSage, which utilizes aggregation functions to generate embeddings for each node in its neighborhood. While these works have significantly advanced GNNs by presenting different architectures, our research focuses on adapting deep mutual learning techniques to enhance the training and performance of existing GNN architectures, rather than proposing new architectures.

**Knowledge Distillation.** Knowledge distillation, proposed by Hinton et al. (2015), has become widely adopted for training compact models under the supervision of larger teacher models. The applicability of KD spans various domains and applications, including model compression Polino et al. (2018), reinforcement learning Rusu et al. (2015), and enhancement of generalization capabilities Tang et al. (2016). In GNNs, KD techniques have been adapted to improve model performance Guo et al. (2023), mitigate negative effects of graph augmentation Wu et al. (2022), and feature transformation Yang et al. (2021). Notable KD techniques for GNNs include TinyGNN by Yan et al. (2020), a framework which enables smaller GNNs to learn local structural knowledge from deeper models, and the method proposed by Deng and Zhang (2021), leveraging multivariate Bernoulli distributions to model graph

---

[1]Our codes can be found in the anonymous link: https://anonymous.4open.science/r/collab-gnn-46DF/

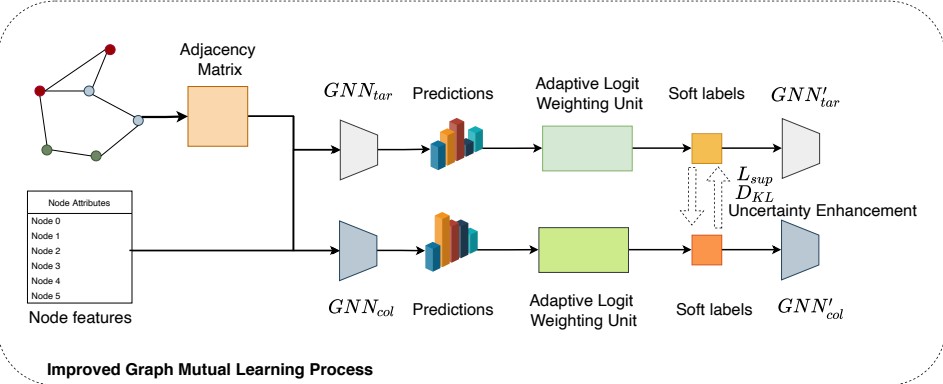

Figure 1: The GML architecture with two GNN models, $GNN_{tar}$ and $GNN_{col}$, training together. The adaptive logit weight unit prioritizes crucial knowledge and the uncertainty enhancement unit penalizes models with low entropy. The process yields two improved models $GNN'_{tar}$ and $GNN'_{col}$.

topology structures for effective knowledge transfer. Similarly, Zhou et al. (2021) presented a strategy for distilling holistic knowledge from attributed graphs through a contrastive learning approach. Zhang et al. (2021) proposed a method to facilitate KD from GNNs to enhance the performance of MLPs, which is beneficial for accelerated inference. While these techniques demonstrate the potential of KD for improving GNN performance, our research aims to investigate a novel approach for KD that leverages mutual learning techniques to enhance transferability and proficiently transfer improved knowledge to an MLP for downstream tasks.

**Collaborative Learning.** Similar works on collaborative learning can be found in the natural language processing (NLP) domain. Liu et al. (2023a) introduce a generator-predictor framework for rationalization, where multiple generators offer varied insights to the predictor to tackle poor correlation and degeneration problems. This framework is characterized by a many-to-one relationship. In contrast, our approach employs peer-based learning across GNNs, utilizing mutual learning and distilling the knowledge into an MLP. The authors in Liu et al. (2024b) used KL-divergence to maximize the separation between target labels and irrelevant parts of the text, proposing the minimal conditional dependence criterion, whereas we employ the method to allow models to distill crucial knowledge from other peers in the same training cohort. Liu et al. (2023b) suggest varying learning rates for individual generators and predictors to address the degeneration problem and encourage improved cooperation. However, controlling through Lipschitz continuity can hinder the adaptive nature of the model and is more challenging to implement. In contrast, we provide a temperature control mechanism that is easier to implement and allows dynamic control over model prediction.

**Deep Mutual Learning.** Deep Mutual Learning (DML), introduced by Zhang et al. (2018), extends the KD framework from its conventional uni-directional transfer to enable bidirectional knowledge exchange between models. Since its introduction, DML has been widely adopted across various domains within DL, including federated learning and Bayesian neural networks Wang et al. (2024a); Liu et al. (2024a); Luo and Zhang (2024). For instance, Wang et al. (2024b) leverage DML for client updates in a study focused on employing heterogeneous model reassembly for personalized federated learning. Similarly, Pham et al. (2024) utilize DML to enhance the performance of Bayesian Neural Networks. In the context of GNNs, Li et al. (2024) adapted DML to multi-modal recommendation tasks, emphasizing collaborative training across uni-modal bipartite user-item graphs. While these applications demonstrate the versatility of DML, our study investigates its applicability for node and graph classification tasks and introduces novel techniques to enhance DML in graph learning.

## 3 METHODOLOGY

**Problem Statement.** Consider a graph $G = (V, E, X)$, where $V$ is the set of $N$ nodes, $E \subseteq V \times V$ are the observed links, and $X \in \mathbb{R}^{N \times D}$ is the attributes matrix. Each node $v_i \in V$ has a $D$-dimensional attribute vector $x_i \in \mathbb{R}^D$. For graph classification tasks, each graph is associated with a label $Y = \{y_i\}_{i=1}^M$, where $y_i \in \{1, 2, 3, \ldots, C\}$, $C$ is the total number of classes, such that for any class $c$, $1 \le c \le C$, and $M$ is the number of graphs. For node classification, $Y = \{y_i\}_{i=1}^N$, where

$y_i \in \{1, 2, 3, \ldots, C\}$, and $N$ is the number of nodes in the graph. Given the posterior probability $p_1$ from a GNN $\theta_1$ for a node $v$, the objective is to improve the generalization performance of $\theta_1$ by using another model $\theta_2$ to provide knowledge in the form of its posterior probability $p_2$.

**Proposed Approach.** Our approach involves collaboratively training a set of untrained shallow GNNs by matching their posterior probabilities, as depicted in Figure 1. This technique aims to enhance the performance of a target model participating in the collaborative training process. Subsequently, we adapt the target model for KD (details of the KD architecture are provided in Appendix A). The untrained GNN cohort comprises models initialized differently and may have different architectures, each featuring a classifier producing a probability distribution over the available classes. We show in Section 3.1 that different random initializations lead to diverse feature representations among the models. Our mutual learning method consists of three unique parts: (1) mutual learning for graph learning, (2) adaptive logit weighting, and (3) uncertainty enhancement. Each component contributes to improving the generalization performance of the target model.

### 3.1 EXPLORING GRAPH NEURAL NETWORK ARCHITECTURES FOR MUTUAL LEARNING

We investigate how different GNN architectures encode features. Different from the analysis of Guo et al. (2023), we also consider the impact of different random initializations on the similarities between the learned representations of similar models. To assess the similarities between layers in different model combinations, we utilize Centered Kernel Alignment (CKA) Kornblith et al. (2019) as our metric. CKA measures the similarity between representations learned by different models, with a higher CKA value indicating greater similarity. We conduct experiments involving training three distinct GNN architectures—3-layer GCN, Graph Attention Network (GAT), and GraphSage—using the Citeseer dataset. For each layer, we compute the average pooling of all embeddings, which serves as the representation for that layer. Figure 2 illustrates the layer-wise similarities among the models.

Initially, we examine the results obtained from using similar model architectures for mutual learning but with different random initializations. Figures 2 (a) and 2 (b) reveal that the similarities between layers $1/2/3$ of two GCN architectures are $0.87/0.18/0.63$, while those between layers $1/2/3$ of two GraphSage architectures are $0.43/0.21/0.19$. Figures 2 (c) and 2 (d) present the similarity between layers of diverse models with varying random initializations. In particular, the results illustrate that the similarities between layers $1/2/3$ of GCN and GAT models are $0.27/0.05/0.44$, and those between layers $1/2/3$ of GCN and GraphSage models are $0.22/0.076/0.046$. These results suggest that GNNs with differing architectures and random initializations yield dissimilar embeddings. In addition, the result shows that when initialized differently, GNNs with the same architectures can diverge in how they encode features in their internal layers.

Leveraging insights from our analysis, we apply our mutual learning technique using various model architectures and random initialization. This approach enables the models to acquire diverse knowledge, thereby enhancing generalization.

### 3.2 GRAPH MUTUAL LEARNING

Our formulation for graph mutual learning involves a collaborative training approach between a cohort of two shallow GNNs to improve their generalization performance. Extension to more than two peers is straightforward and is given in Appendix B. Given a graph dataset $G$ for node classification, each model $\theta_j$ predicts the probability of the $c$-th class for node $v$ using the softmax function with temperature scaling:

$$p_j^c = \frac{\exp(z_j^{v,c}/T_v)}{\sum_{c=1}^{C} \exp(z_j^{v,c}/T_v)} \tag{1}$$

Here, $z_j^{v,c}$ represents the logits for class $c$ produced by model $\theta_j$, and $T_v$ is the temperature parameter for node $v$ used to soften the logits, controlling the sharpness of the probability distribution.

In mutual learning, one GNN model, denoted as the target model $\theta_{tar}$, collaborates with another peer model $\theta_{col}$ by leveraging its posterior probability distribution $p_{col}$ as shared knowledge to improve its generalization. Each model in the cohort has a local supervision loss $L_{sup}$ between the predicted logits and the correct labels. Mutual learning aims to align the probability distributions of the two

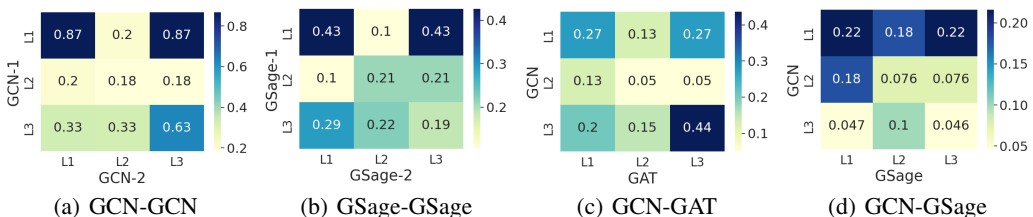

Figure 2: Centered kernel alignment similarity between model layers 1, 2, and 3. (a) and (b) show the similarity between models of the same architecture but different initializations. (c) and (d) show the similarity between models of different architectures and different initializations.

models, encouraging them to learn from each other's predictions. This alignment is achieved through the Kullback Leibler (KL) Divergence loss. Thus, the overall loss of models $\theta_{tar}$ and $\theta_{col}$ is given as:

$$L_{tar} = L_{sup_{tar}} + D_{KL}(p_{col}||p_{tar}) \tag{2}$$

$$L_{col} = L_{sup_{col}} + D_{KL}(p_{tar}||p_{col}) \tag{3}$$

where $L_{sup_{tar}}$ and $L_{sup_{col}}$ represent the local supervision losses for the target and collaborative model, respectively. The KL divergence terms for the target and collaborative peer, $D_{KL}(p_{col}||p_{tar})$ and $D_{KL}(p_{tar}||p_{col})$, measure the discrepancy between the probability distributions of the two models, encouraging them to converge towards similar predictions. For our setting, we use the cross-entropy loss as the local supervision loss for each model, ensuring that they learn to predict the correct class labels for the graph nodes.

### 3.3 ADAPTIVE LOGIT WEIGHTING

The adaptive logit weighting module is designed to prioritize shared knowledge during the mutual learning process between two shallow GNNs. This module consists of two learnable variables, $\chi_j$ and $\phi_j$, where $\chi_j \in \mathbb{R}^{N \times h}$ and $\phi_j \in \mathbb{R}^{h \times C}$. Given the prediction probabilities $p_j$ for all nodes $V$, the module calculates the negative entropy of the logits, denoted as $H(p_j) \in \mathbb{R}^{N \times 1}$, to measure the model's confidence. This entropy information is then used to compute an adaptive weight vector $W_j^c$ for each class $c$, ensuring that more important logits receive higher weights. The adaptive weight vector $W_j^c$ of the $c$-th class is computed as:

$$W_j^c = \frac{exp(\sigma_j^c)}{\sum_{c=1}^{C} exp(\sigma_j^c)} \tag{4}$$

where $\sigma_j^c$ represents the importance score for class $c$ obtained from the negative entropy $H(p_j)$ and the learned parameters $\chi_j$ and $\phi_j$. $\sigma_j$ is given as $H(p_j)^T \chi_j \phi_j \in \mathbb{R}^{1 \times C}$ and $\sigma_j^c \in \sigma_j$.

Subsequently, for each node $v$, the prediction probabilities $p_j \in \mathbb{R}^{1 \times C}$ are adjusted based on the adaptive weight vector $W_j \in \mathbb{R}^{1 \times C}$ using the Hadamard product:

$$p\prime_j = p_j \cdot W_j \tag{5}$$

The adaptive logit weighting module is trained jointly with the participating models by minimizing the loss function, which includes the KL divergence between the adjusted prediction probabilities of the target and collaborative models, along with regularization terms for the learnable variables:

$$L_{tar} = L_{sup_{tar}} + D_{KL}(p\prime_{col}||p\prime_{tar}) + \beta(||\chi_{tar}|| + ||\phi_{tar}||) \tag{6}$$

$$L_{col} = L_{sup_{col}} + D_{KL}(p\prime_{tar}||p\prime_{col}) + \beta(||\chi_{col}|| + ||\phi_{col}||) \tag{7}$$

Here $||\chi_{tar}||$ and $||\psi_{tar}||$ are the $L_1$ norms of the learnable variables for the target model. $||\chi_{col}||$ and $||\psi_{col}||$ are for the collaborating peer. $\beta$ is a hyperparameter used to balance the regularization terms, controlling the impact of the $L_1$ norms of the learnable variables.

### 3.4 ENHANCING UNCERTAINTY

Ensuring appropriate uncertainty in machine learning models is crucial for their generalizability and adaptability to various real-world scenarios. In our mutual learning framework between two models $\theta_1$ and $\theta_2$, we aim to enhance uncertainty to prevent overfitting and improve adaptability. We begin by examining the KL divergence that matches the posterior probabilities $p_1$ and $p_2$ between the two models for a training example:

$$D_{KL}(p_2||p_1) = \sum_{c=1}^{C} p_2^c \log \frac{p_2^c}{p_1^c} \qquad (8)$$

Expanding Equation 8, as shown in Appendix C, reveals that the equation can be decomposed into a negative entropy and a cross-entropy term.

During optimization, we aim to minimize $D_{KL}(p_2||p_1)$ with respect to $z_1^{v,c}$ (from Eq. 1), the logits of $\theta_1$. According to Hinton et al. (2015), optimizing this divergence yields:

$$\frac{\partial D_{KL}(p_2||p_1)}{\partial z_1^v} = \tau(p_1 - p_2) \qquad (9)$$

where $\tau$ is a temperature scaling parameter. If the probability distributions perfectly match, no knowledge transfer occurs between the models.

While minimizing the KL divergence implicitly considers the entropy of the distribution, it primarily focuses on cross-entropy. When the student's predicted distribution matches the teacher's distribution and the ground-truth logit is significantly higher than the other logits, the student can become overconfident. This overconfidence leads to the student assigning nearly all probability to a single class, resulting in overfitting and reduced adaptability Szegedy et al. (2016). To address this, we introduce a confidence penalty term to the loss functions of each model. This penalty term serves as a regularization factor, discouraging peaked distributions by maintaining appropriate uncertainty levels during training Pereyra et al. (2017). Specifically, we incorporate $H(p_{tar}|v)$ and $H(p_{col}|v)$, denoting the negative entropy of the predicted probabilities for a given training example $v$. Where $\gamma$ is a hyperparameter that balances the contribution of the confidence penalty terms, the loss of the participating models is:

$$L_{tar} = L_{sup_{tar}} + D_{KL}(p_{col}||p_{tar}) - \gamma H(p_{tar}|v) \qquad (10)$$
$$L_{col} = L_{sup_{col}} + D_{KL}(p_{tar}||p_{col}) - \gamma H(p_{col}|v) \qquad (11)$$

## 4 EXPERIMENTS

### 4.1 EXPERIMENTAL SETUP

The following outlines our experimental setup, including datasets, hardware specifications, and the baseline models we employed for GML.

**Datasets** We evaluate the performance of our GML approach using three widely-used datasets for both node classification and graph classification tasks Sen et al. (2008); Namata et al. (2012); Hu et al. (2020); Borgwardt et al. (2005). For node classification, we employ the Cora, Citeseer, and PubMed datasets. For graph classification, we utilize the PROTEINS dataset along with two Open Graph Benchmark (OGB) datasets Hu et al. (2020): Ogbg-molbace and Ogbg-molbbbp. Additionally, we assess the performance of our approach adapted for KD using five datasets for node classification Sen et al. (2008); Namata et al. (2012); Shchur et al. (2018): Cora, Citeseer, PubMed, Amazon Computers, and Amazon Photo. Detailed information about these datasets is provided in Appendix D.

**Models.** We use the GCN Kipf and Welling (2016), GAT Veličković et al. (2017), and Graph-Sage Hamilton et al. (2017) as our baseline GNN models. GCN introduces a fundamental approach to graph representation learning by aggregating information from neighboring nodes through graph convolutions. GAT incorporates attention mechanisms that dynamically compute attention coefficients

based on node features, allowing the model to focus on informative neighbors. GraphSage aggregates information from sampled neighboring nodes using different aggregation functions, enabling the model to capture diverse neighborhood information. We use three combinations for the diverse architectural design and also evaluate our approach using architectural design with the same type of GNN models. In Appendix E, we provide further details about the architectures of these models.

**Evaluation.** In our experiments, we evaluate all approaches using accuracy as the primary metric. Due to the complexity of graph classification tasks, we report the average accuracy with the standard deviation after five training iterations for each experiment involving graph classification tasks with mutual learning. For other experiments, we report the average accuracy with the standard deviation derived from ten training iterations.

## 4.2 Performance Evaluation

We evaluate GML's potential to enhance the performance of baseline GNNs through collaborative training. We then investigate whether these enhancements can be effectively leveraged to improve the performance of GML in graph learning tasks. We evaluate three distinct model combinations to assess the effectiveness of GML across a diverse range of architectures and initialization schemes: GraphSage-GCN, GAT-GraphSage, and GCN-GAT. For GraphSage-GCN and GAT-GraphSage architectures, we present results with GraphSage as the target model, while for GCN-GAT, we focus on GCN as the target model. Our experimental findings, detailed in Table 1, demonstrate the efficacy of GML in improving the performance of baseline GNN models across various scenarios. For instance, consider the GAT-GraphSage architecture with GraphSage as the target model on the Cora dataset. Without GML, the accuracy stands at $86.58\%$, whereas with GML, it improves to $88.55\%$. Our results consistently show that integrating GML with our enhancements leads to notable performance improvements over baseline GNN models. For example, employing the GAT-GraphSage combination with GraphSage as the target model results in an accuracy increase from $69.64\%$ to $70.36\%$ on the PROTEINS dataset with the introduction of the confidence penalty technique. Similarly, using the GraphSage-GCN combination with GraphSage as the target model on the Ogbg-molbbp dataset shows promising results. The initial accuracy of the baseline GNN model improves from $84.20\%$ to $85.38\%$ with the integration of the adaptive logit weighting technique. These findings underscore the significance of employing GML with appropriate enhancement techniques to improve the performance of shallow GNN models.

| Models | Methods | Node Classification | | | Graph Classification | | |
|---|---|---|---|---|---|---|---|
| | | Cora | Citeseer | PubMed | Ogbg-molbace | Ogbg-molbbbp | PROTEINS |
| GraphSage-GCN-S | Ind | $86.58 \pm 0.68$ | $76.57 \pm 1.24$ | $89.03 \pm 0.50$ | $77.43 \pm 1.90$ | $84.20 \pm 0.63$ | $69.64 \pm 1.25$ |
| | GML | $87.00 \pm 0.52$ | $\mathbf{77.80 \pm 0.51}$ | $89.66 \pm 0.34$ | $78.32 \pm 0.94$ | $84.79 \pm 0.85$ | $69.28 \pm 1.54$ |
| | GML-Co | $87.32 \pm 0.50$ | $75.99 \pm 0.66$ | $\mathbf{90.19 \pm 0.30}$ | $\mathbf{79.03 \pm 1.67}$ | $84.07 \pm 1.1$ | $69.76 \pm 0.79$ |
| | GML-W | $87.49 \pm 0.40$ | $76.93 \pm 0.50$ | $89.17 \pm 0.25$ | $77.79 \pm 0.37$ | $\mathbf{85.38 \pm 0.94}$ | $70.24 \pm 0.91$ |
| | GML-C | $\mathbf{88.69 \pm 0.38}$ | $76.13 \pm 0.46$ | $90.17 \pm 0.13$ | $78.23 \pm 1.23$ | $85.05 \pm 0.50$ | $\mathbf{70.96 \pm 1.08}$ |
| GAT-GraphSage-S | Ind | $86.58 \pm 0.68$ | $76.57 \pm 1.24$ | $89.03 \pm 0.5$ | $77.43 \pm 1.90$ | $84.20 \pm 0.63$ | $69.64 \pm 1.25$ |
| | GML | $88.55 \pm 0.24$ | $77.37 \pm 0.59$ | $89.7 \pm 0.24$ | $78.23 \pm 1.34$ | $84.07 \pm 1.15$ | $69.4 \pm 0.99$ |
| | GML-Co | $87.88 \pm 0.68$ | $76.61 \pm 0.62$ | $90.13 \pm 0.30$ | $78.58 \pm 1.55$ | $\mathbf{84.72 \pm 1.2}$ | $69.64 \pm 1.45$ |
| | GML-W | $87.36 \pm 2.79$ | $76.45 \pm 0.75$ | $89.20 \pm 0.20$ | $\mathbf{78.76 \pm 1.13}$ | $84.39 \pm 0.82$ | $70.36 \pm 0.79$ |
| | GML-C | $\mathbf{88.57 \pm 0.41}$ | $\mathbf{77.45 \pm 0.6}$ | $90.24 \pm 0.23$ | $78.05 \pm 1.27$ | $84.07 \pm 2.03$ | $\mathbf{70.36 \pm 1.44}$ |
| GCN-GAT-C | Ind | $88.87 \pm 0.10$ | $76.55 \pm 0.21$ | $87.20 \pm 0.05$ | $75.93 \pm 1.64$ | $84.52 \pm 0.27$ | $71.45 \pm 1.32$ |
| | GML | $89.24 \pm 0.39$ | $76.71 \pm 0.23$ | $87.20 \pm 0.18$ | $73.54 \pm 1.48$ | $84.26 \pm 0.9$ | $71.20 \pm 1.16$ |
| | GML-Co | $89.06 \pm 0.24$ | $\mathbf{76.73 \pm 0.27}$ | $87.33 \pm 0.10$ | $75.22 \pm 2.00$ | $84.26 \pm 0.52$ | $71.20 \pm 1.62$ |
| | GML-W | $89.24 \pm 0.26$ | $76.63 \pm 0.17$ | $87.15 \pm 0.10$ | $\mathbf{76.48 \pm 0.85}$ | $\mathbf{84.59 \pm 0.52}$ | $71.08 \pm 0.60$ |
| | GML-C | $\mathbf{89.26 \pm 0.31}$ | $76.61 \pm 0.23$ | $\mathbf{87.45 \pm 0.14}$ | $74.87 \pm 1.61$ | $84.33 \pm 0.82$ | $\mathbf{71.81 \pm 0.27}$ |

Table 1: Classification performance of mutual learning for node and graph classification. Ind is the performance of the target model without any mutual learning. S and C represent GraphSage and GCN as target models, respectively. Thus, GraphSage-GCN-S denotes GML with GraphSage and GCN, using GraphSage as the target model. GML-Co represents the performance of the target model with adaptive logit weighting and uncertainty enhancement. GML-W denotes the performance with only adaptive logit weighting and GML-C represents the performance with uncertainty enhancement.

### 4.3 BEYOND GRAPHS TO GRAPH-LESS NEURAL NETWORKS

Beyond GML, we investigate whether the improvements achieved through GML can be transferred to a simple MLP via KD to satisfy the requirement for faster inference in industrial settings. The distilled MLP is referred to as a graph-less neural network Zhang et al. (2021). Our analysis consists of diverse architecture settings, including GraphSage-GCN, GAT-GraphSage, and GCN-GAT architectures. For the KD process, we maintain GraphSage as the teacher model from GraphSage-GCN and GAT-GraphSage architectures, while GCN serves as the teacher obtained from the GCN-GAT architecture. Table 2 presents the results of our experiment. We found that KD improves the performance of individual MLPs by leveraging deep mutual learning with our specific enhancements (adaptive logit weighting or confidence penalty). For example, using GraphSage as the teacher model from a GraphSage-GCN combination increases the accuracy of the MLP on the Cora dataset from $70.10\%$ to $87.66\%$ while using the adaptive logit weighting unit. Through KD, the teacher model acquires more generalizable and transferable knowledge, enabling efficient training of student MLP models that can rival the baseline teacher model in competitiveness.

| Models | Methods | Cora | Citeseer | PubMed | A-Computers | A-photo |
|---|---|---|---|---|---|---|
| | Ind-MLP | $70.10 \pm 1.12$ | $67.88 \pm 0.53$ | $84.25 \pm 0.75$ | $77.59 \pm 0.64$ | $87.42 \pm 0.75$ |
| GraphSage-GCN-S | KD without GML | $86.01 \pm 0.19$ | $75.11 \pm 0.08$ | $88.98 \pm 0.14$ | $89.03 \pm 0.31$ | $94.87 \pm 0.09$ |
| | KD + GML | $86.18 \pm 0.24$ | $76.01 \pm 0.16$ | $89.45 \pm 0.17$ | $\mathbf{89.48 \pm 0.30}$ | $95.02 \pm 0.13$ |
| | KD + GML-Co | $87.49 \pm 0.28$ | $75.81 \pm 0.30$ | $\mathbf{89.53 \pm 0.20}$ | $89.44 \pm 0.27$ | $94.56 \pm 0.16$ |
| | KD + GML-W | $\mathbf{87.66 \pm 0.24}$ | $\mathbf{76.27 \pm 0.10}$ | $88.46 \pm 0.22$ | $89.46 \pm 0.26$ | $94.08 \pm 0.13$ |
| | KD + GML-C | $86.03 \pm 0.23$ | $75.71 \pm 0.26$ | $89.24 \pm 0.28$ | $89.04 \pm 0.20$ | $\mathbf{95.18 \pm 0.15}$ |
| GAT-GraphSage-S | KD without GML | $86.01 \pm 0.19$ | $75.11 \pm 0.08$ | $88.98 \pm 0.14$ | $89.03 \pm 0.31$ | $94.87 \pm 0.09$ |
| | KD + GML | $85.67 \pm 0.10$ | $76.55 \pm 0.19$ | $89.58 \pm 0.27$ | $90.04 \pm 0.23$ | $94.95 \pm 0.20$ |
| | KD + GML-Co | $\mathbf{87.44 \pm 0.16}$ | $75.97 \pm 0.26$ | $\mathbf{89.88 \pm 0.23}$ | $89.47 \pm 0.20$ | $94.43 \pm 0.09$ |
| | KD + GML-W | $85.99 \pm 0.32$ | $76.27 \pm 0.01$ | $89.36 \pm 0.25$ | $88.96 \pm 0.20$ | $94.80 \pm 0.19$ |
| | KD + GML-C | $86.33 \pm 0.26$ | $\mathbf{76.57 \pm 0.45}$ | $89.42 \pm 0.21$ | $\mathbf{90.10 \pm 0.25}$ | $\mathbf{95.43 \pm 0.10}$ |
| GCN-GAT-C | KD without GML | $88.52 \pm 0.51$ | $78.92 \pm 0.23$ | $88.45 \pm 0.15$ | $77.04 \pm 0.36$ | $90.17 \pm 0.52$ |
| | KD + GML | $87.91 \pm 0.22$ | $79.00 \pm 0.16$ | $\mathbf{88.55 \pm 0.23}$ | $77.54 \pm 0.40$ | $90.68 \pm 0.41$ |
| | KD + GML-Co | $88.42 \pm 0.23$ | $78.82 \pm 0.38$ | $88.29 \pm 0.39$ | $76.36 \pm 0.64$ | $90.21 \pm 0.40$ |
| | KD + GML-W | $88.32 \pm 0.28$ | $78.42 \pm 0.30$ | $88.43 \pm 0.20$ | $77.36 \pm 0.34$ | $\mathbf{90.72 \pm 0.37}$ |
| | KD + GML-C | $\mathbf{88.72 \pm 0.19}$ | $\mathbf{79.48 \pm 0.22}$ | $88.49 \pm 0.41$ | $\mathbf{77.82 \pm 0.37}$ | $90.62 \pm 0.29$ |

Table 2: Performance of MLP with KD using the target model. Ind is the performance of the MLP without any mutual learning. We use models with different architectures for mutual learning.

### 4.4 DOES KD WORK WITH A TARGET MODEL THAT LEARNS FROM A PEER WITH SIMILAR ARCHITECTURE?

We explore whether KD remains effective when applied to pairs of GNNs with identical architectural settings. Specifically, we consider three possible pairs: GraphSage-GraphSage, GAT-GAT, and GCN-GCN. Our assessment focuses on determining if GML's performance and proposed enhancements extend to scenarios where the target and peer models share the same architecture. The results of our evaluation are presented in Table 3. They show that GNNs with identical architectures can indeed exchange essential knowledge to enhance their performance. The proposed enhancements also improve the performance of GNNs with similar architectures, facilitating better generalization and emphasizing critical knowledge exchange during mutual learning. For example, using GAT as the teacher model from a GAT-GAT combination increases the accuracy of the MLP on the Citeseer dataset from $77.69\%$ to $80.56\%$ while using the adaptive logit weighting unit. This observation aligns with our initial investigation into GNN embeddings, revealing that even with identical architectures, GNNs may encode distinct embeddings when initialized with different random seeds.

### 4.5 ABLATION STUDIES

**Does mutual learning facilitate knowledge distillation?** Tables 2 and 3 demonstrate the enhanced performance of the vanilla MLP model following KD using the teacher model derived from the GML process. Notably, the improvements in the MLP model stem from the superior performance of the teacher model, which has benefited from prior enhancement through GML. Figure 3 shows the outcomes of our experiments across diverse architectures, confirming the correlation between

| Models | Methods | Cora | Citeseer | PubMed | A-Computers | A-photo |
|---|---|---|---|---|---|---|
| | Ind-MLP | 69.14 ± 1.28 | 68.28 ± 0.85 | 85.29 ± 0.49 | 77.1 ± 0.68 | 86.83 ± 0.60 |
| GraphSage-GraphSage | KD without GML | 89.06 ± 0.13 | 76.73 ± 0.26 | 89.75 ± 0.19 | 88.66 ± 0.38 | 95.3 ± 0.22 |
| | KD + GML | 88.65 ± 0.80 | 76.55 ± 0.16 | 90.22 ± 0.20 | 88.79 ± 0.19 | 94.73 ± 0.13 |
| | KD + GML-Co | 88.99 ± 1.7 | 76.11 ± 0.16 | **90.39 ± 0.28** | 88.63 ± 0.15 | **95.35 ± 0.29** |
| | KD + GML-W | **89.63 ± 0.14** | **78.06 ± 0.14** | 89.63 ± 0.16 | **89.04 ± 0.17** | 95.17 ± 0.15 |
| | KD + GML-C | 89.33 ± 0.17 | 76.31 ± 0.32 | 90.24 ± 0.17 | 88.39 ± 0.38 | 94.38 ± 0.27 |
| GAT-GAT | KD without GML | 87.98 ± 0.34 | 77.69 ± 0.12 | 88.72 ± 0.19 | 85.52 ± 0.19 | 90.01 ± 0.25 |
| | KD + GML | **88.18 ± 0.33** | 79.92 ± 0.16 | 88.92 ± 0.31 | 86.53 ± 0.35 | 90.63 ± 0.39 |
| | KD + GML-Co | 86.35 ± 0.33 | 80.20 ± 0.28 | **89.17 ± 0.28** | **86.67 ± 0.35** | **92.39 ± 0.23** |
| | KD + GML-W | 87.04 ± 0.24 | **80.56 ± 0.25** | 88.82 ± 0.21 | 84.78 ± 0.31 | 90.43 ± 0.24 |
| | KD + GML-C | 88.15 ± 0.36 | 80.12 ± 0.32 | 89.12 ± 0.32 | 85.81 ± 0.36 | 91.91 ± 0.41 |
| GCN-GCN | KD without GML | 89.04 ± 0.29 | 79.90 ± 0.14 | 89.37 ± 0.22 | 77.20 ± 0.44 | 89.09 ± 0.41 |
| | KD + GML | 89.14 ± 0.18 | 80.26 ± 0.14 | 89.37 ± 0.25 | 77.48 ± 0.47 | 89.22 ± 0.41 |
| | KD + GML-Co | **89.29 ± 0.21** | 80.28 ± 0.27 | **89.51 ± 0.35** | 78.22 ± 0.47 | 89.16 ± 0.68 |
| | KD + GML-W | 89.24 ± 0.17 | 80.14 ± 0.24 | 89.29 ± 0.14 | 75.46 ± 0.71 | **89.54 ± 0.46** |
| | KD + GML-C | 89.19 ± 0.18 | **80.54 ± 0.36** | 89.43 ± 0.21 | **78.73 ± 0.71** | 89.53 ± 0.68 |

Table 3: Performance of MLP with KD using the target model. Ind is the performance of the MLP without any mutual learning. We use models with same architecture for mutual learning.

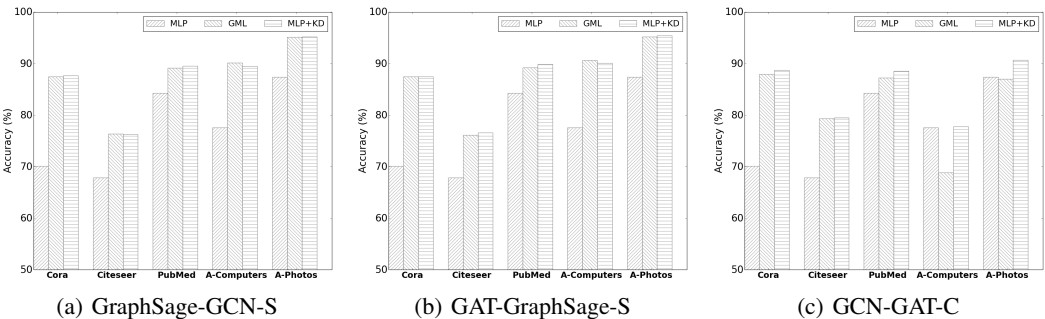

(a) GraphSage-GCN-S  (b) GAT-GraphSage-S  (c) GCN-GAT-C

Figure 3: Results of KD with the best performing GML technique. We show the difference in the performance of KD with the MLP without KD as influenced by GML.

the improvement in the MLP model and the enhanced performance of its corresponding teacher model. However, the extent of performance enhancement varies across datasets and architectures. For example, the smallest performance gain is observed in the A-Computers dataset employing the GCN-GAT architecture with GCN as the teacher model. This observation underscores the dependency of the vanilla MLP model's improvement on the quality of the teacher model and its comparative performance against the student MLP.

**Impact of hyperparameters $\gamma$, $\beta$ and temperature $T_v$.** We conducted experiments using the GraphSage-GCN combination, with GraphSage as the target, to analyze the sensitivity of the parameters $\beta$, $\gamma$, and $T_v$. In this analysis, we systematically varied one parameter at a time while keeping the others constant. The results, as depicted in Figure 4 indicate that the parameter $\gamma$ remains relatively stable across a wide range of values, with a noticeable decrease in accuracy observed for larger values of $\gamma$. Conversely, the parameter $\beta$ demonstrated the highest accuracy with larger values. As for $T_v$, the highest accuracy was achieved when its value was 1.0, with accuracy decreasing as $T_v$ was increased to 10. These findings suggest that all three parameters exhibit stability over large intervals.

**Does expanding the cohort impact performance?** We evaluate the performance of our GML technique by aggregating predictions from models that offer complementary perspectives. Initially, we assess GML's performance using models of identical architecture that are initialized with different random seeds. We then extend the evaluation to include models with distinct architectures, randomly selected from the pool of models in Section 4.2. The results are summarized in Figure 5. Our experiment demonstrates a general improvement in performance as the number of models in the cohort increases, indicating that our approach scales effectively with an increasing number of models

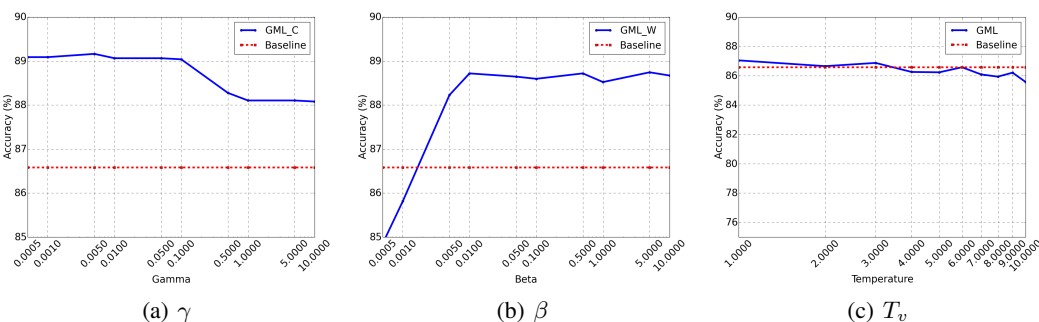

(a) $\gamma$        (b) $\beta$        (c) $T_v$

Figure 4: Sensitivity analysis of the balancing parameters $\gamma$, $\beta$ and the temperature $T_v$.

in the cohort. With greater hardware resources, this scalability can be further exploited through parallelization, enabling larger cohorts to enhance GML performance.

**How does GML compare with ensemble learning without collaboration?** We apply the same selection methods from the expanding cohort experiment to compare the performance of GML with standard ensemble techniques. As shown in Figure 5, our results indicate that traditional ensemble methods consistently outperform single-model predictions. Specifically, for ensemble sizes greater than five, performance improves notably when the ensemble consists of diverse models. When we adapt our approach to ensemble techniques, GML demonstrates superior performance compared to standard ensemble methods. Moreover, the inclusion of diverse models in the ensemble enhances predictive accuracy for ensemble sizes where $n > 5$ (with $n$ representing the number of models). This suggests that GML effectively capitalizes on model diversity, leveraging complementary knowledge to boost overall performance.

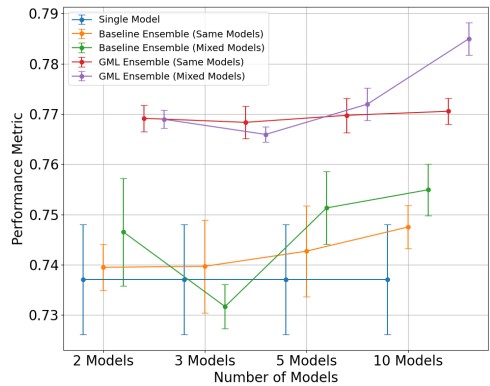

Figure 5: Comparison of GML ensemble method with deep ensembles.

**Effect of noisy graph structures on GML.** In previous research, Bechler-Speicher et al. (2023) noted that GNNs often overfit to graph structures, especially in situations where disregarding noisy structures could lead to better performance. We observe that GML can reduce this overfitting tendency in GNNs. The detailed experimental setup and results are provided in Appendix I.

**Impact of adaptive logit weighting unit and the confidence penalty mechanism** To evaluate the contributions of the logit weighting unit and the confidence penalty mechanism to improving GML performance, we conducted additional experiments focused on node classification using the largest dataset in our paper for this task, PubMed. Our result is in Appendix J, with a sanity check in Appendix K showing the improvements are not due to random fluctuations.

## 5   CONCLUSION AND FUTURE WORK

In this paper, we introduced Graph Mutual Learning (GML), a novel approach that leverages deep mutual learning techniques to enhance GNNs. We augmented the mutual learning process with two key techniques: adaptive logit weighting and a confidence penalty term, which proved effective in transferring crucial knowledge between collaborating peers and promoting entropy for improved generalization. Furthermore, we adapt our approach for KD, demonstrating that knowledge acquired during the mutual learning process can be effectively transferred to a student model for downstream tasks. Extensive experiments on node and graph classification datasets empirically demonstrate that our approach can enhance shallow GNN models through online distillation techniques. Future work will focus on evaluating the robustness of GML against noisy or adversarial data and developing novel techniques to enhance its scalability, particularly for large-scale graph datasets.

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

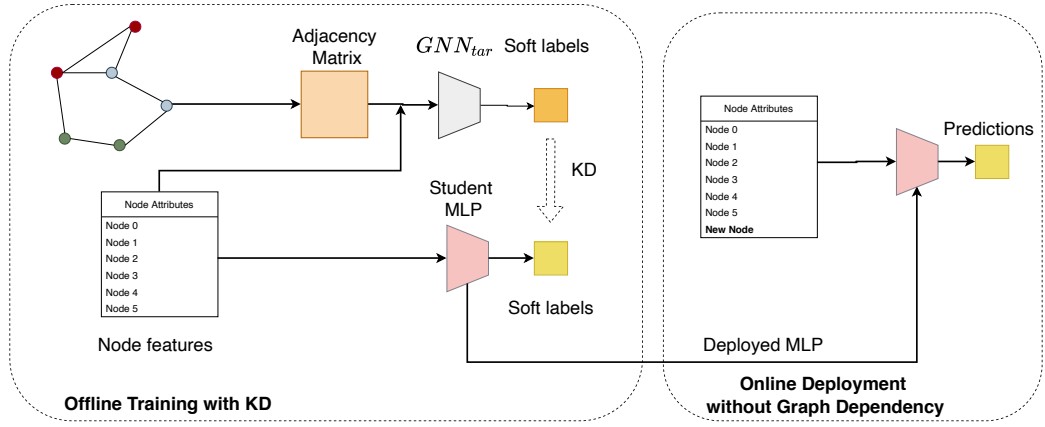

Figure 6: Architecture of the KD and Deployment Processes Between GNN and MLP.

## A  KNOWLEDGE DISTILLATION (KD) ARCHHITECTURE

KD was introduced by Hinton et al. Hinton et al. (2015), where a student learns from a larger teacher model. Zhang et al. Zhang et al. (2021) extended this idea to GNNs, which generate soft targets that are used to train a student MLP. Given that the soft target from the teacher model is $z_v$ and the prediction of the student is $\hat{y}_v$, the loss function of the student is given by:

$$L_{student} = L_{sup} + D_{KL}(z_v||\hat{y}_v) \tag{12}$$

where $L_{sup}$ is the supervision loss and $D_{KL}(z_v||\hat{y}_v)$ is the KL-divergence between the teacher and student predictions.

In Figure 6, we present the architecture used to train and deploy the MLP. Initially, the MLP undergoes a KD process, learning from a pre-trained GNN in an offline distillation phase. Here, the MLP is trained using node features but benefits from a more robust GNN trained with node features and graph topology information. Following the offline KD, the MLP is deployed online for faster inference, utilizing only the features of new nodes.

## B  EXTENSION OF MUTUAL LEARNING

For a two-peer cohort, our mutual learning loss functions are given as:

$$L_{tar} = L_{sup_{tar}} + D_{KL}(p_{col}||p_{tar}) - \gamma H(p_{tar}|v) \tag{13}$$

$$L_{col} = L_{sup_{col}} + D_{KL}(p_{tar}||p_{col}) - \gamma H(p_{col}|v) \tag{14}$$

Similar to the original deep mutual learning approach Zhang et al. (2018), we can extend it to a cohort of $K$ peers, where the target model takes the average of its KL divergence with the other $K-1$ peers as follows:

$$L_{tar} = L_{sup_{tar}} + \frac{1}{K-1}\sum_{k=1,k\neq tar}^{K-1} D_{KL}(p_k||p_{tar}) - \gamma H(p_{tar}|v) \tag{15}$$

$$L_{col} = L_{sup_{col}} + \frac{1}{K-1}\sum_{k=1,k\neq col}^{K-1} D_{KL}(p_k||p_{col}) - \gamma H(p_{col}|v) \tag{16}$$

## C  EXPANSION OF KL-DIVERGENCE

$$D_{KL}(p_2||p_1) = \sum_{c=1}^{C} p_2^c \log \frac{p_2^c}{p_1^c}$$

$$D_{KL}(p_2||p_1) = \sum_{c=1}^{C} p_2^c (\log p_2^c - \log p_1^c) \tag{17}$$

$$D_{KL}(p_2||p_1) = \sum_{c=1}^{C} p_2^c \log p_2^c - \sum_{c=1}^{C} p_2^c \log p_1^c$$

The first term of the equation is the negative entropy term while the second term is the cross entropy.

## D  DATASET DETAILS

We employ a total of eight datasets in our experiments. Specifically, for the tasks involving mutual learning for node and graph classification, we use three datasets each. Additionally, for KD adaptation, we focus on five node classification datasets. For all node classification tasks, we employ a split ratio of $0.70/0.15/0.15$ for training/validation/testing sets. For graph classification tasks, we utilize a split ratio of $0.75/0.10/0.15$ for the training/validation/testing sets.

**Node Classification Datasets:**

**Cora Dataset.** The Cora dataset Sen et al. (2008) comprises 2708 scientific papers categorized into seven classes, with a citation network containing 5429 connections. Each paper is represented by a binary word vector denoting the presence or absence of each term from a dictionary of 1433 unique words.

**Citeseer Dataset.** The CiteSeer dataset Sen et al. (2008) comprises 3,312 scientific papers categorized into six classes. Within this dataset, there exists a citation network containing 4,732 links. Each paper is represented by a binary word vector indicating whether a particular word from a dictionary of 3,703 unique words is present (1) or absent (0).

**PubMed Dataset.** The PubMed dataset Namata et al. (2012) encompasses 19,717 scientific publications sourced from the PubMed database, focusing on diabetes, and classified into one of three categories. Within this dataset, there exists a citation network comprising 44,338 links. Each publication is represented by a TF/IDF weighted word vector derived from a dictionary containing 500 distinct words.

**Amazon Computers.** Amazon Computers (A-Computers) Shchur et al. (2018) represents goods as nodes and frequent co-purchases as edges to classify goods into their respective product categories using bag-of-words features extracted from product reviews. This dataset consists of 13,752 nodes,491,722 edges, and 767 features with 10 classes.

**Amazon Photo.** Amazon photo (A-Photo) Shchur et al. (2018) represents goods as nodes and frequent co-purchases as edges to classify goods into their respective product categories using bag-of-words features extracted from product reviews. This dataset consists of 7,650 nodes,238,162 edges, and 745 features with 8 classes.

**Graph Classification Datasets:**

**Ogbg-molbace Dataset.** Th molbace dataset Hu et al. (2020) is from the Open Graph Benchmark (OGB) for the task of graph property prediction. The dataset consists of 1,513 graphs with average nodes of 34.1 and average edges of 36.9. The dataset is provided for binary class prediction tasks.

**Ogbg-molbbbp Dataset.** Similar to the molbace dataset, the molbbbp dataset Hu et al. (2020) is from the Open Graph Benchmark (OGB) for the task of graph property prediction. The dataset consists of 2,049 graphs with 24.1 average nodes and 26.0 average edges. The dataset is also provided for binary class prediction tasks.

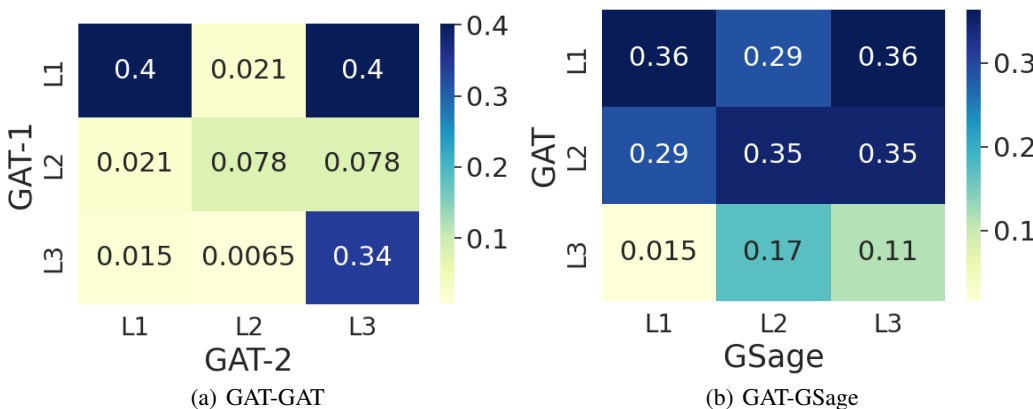

(a) GAT-GAT            (b) GAT-GSage

Figure 7: CKA Similarity Between Layers of Models. L1, L2, and L3 are layers 1, 2, and 3, respectively. (a) shows the similarity between models of the same architecture but different initializations. (b) shows the similarity between models of different architectures and initializations.

**PROTEINS Dataset.** This dataset Borgwardt et al. (2005) was derived from the work of Dobson and Doig Dobson and Doig (2003) and consists of proteins classified as enzymes or non-enzymes. In the PROTEINS dataset, amino acids are represented as nodes and an edge represents the spatial proximity between the nodes. The dataset consists of 1,113 graphs with approximately 39.1 nodes and 145.6 edges. Each node has 3 features and the graph is classified into 1 of 2 available classes.

## E    IMPLEMENTATION DETAILS

We use the Adam optimizer Kingma and Ba (2014) for optimization with a weight decay of $5 \times 10^{-4}$. We designed a 2-layer GCN and GAT and a 3-layer GraphSage. For GCN and GraphSage, we use the ReLU activation for function and the ELU Clevert et al. (2015) activation for GAT. We set the hidden dimensions for node classification tasks to 64, 64, and 8 for the GCN, GraphSage, and GAT models, respectively. The dimensions of $\chi_j$ and $\phi_j$ are set to 64. In GAT, we use 4 attention heads across all node classification datasets. or graph mutual learning experiments, we set $\gamma$ and $\beta$ as 1 for the Citeseer and PubMed datasets, and $\gamma$ as 0.01 for Cora. We set $\gamma = 1$ and $\beta = 1$ for A-Computers during knowledge distillation. For A-Photo, $\gamma$ and $\beta$ were set to 0.1 and 0.01, respectively. The early stopping patience threshold was set to 1500 in node classification experiments.

For graph classification tasks, our models included a read-out layer with a final classifier. We used hidden dimensions of 16, 16, and 8 for GCN, GraphSage, and GAT models. The dimensions of $\chi_j$ and $\phi_j$ are set to 16. In mutual learning experiments for graph classification, we set $T_v$ to 6 and $\gamma$ as 1. The early stopping patience threshold for graph classification experiments was set to 200.

**Hardware Details.** We run all experiments on a single NVIDIA RTX A6000 GPU with 64GB RAM.

**Software Details.** We performed all experiments on a computer with an operating system Ubuntu (version 18.04.6 LTS). We implemented our models using PyTorch Paszke et al. (2017) and Pytorch Geometric Fey and Lenssen (2019).

## F    ADDITIONAL RESULTS ON THE EXPLORATION OF GRAPH NEURAL NETWORK ARCHITECTURES FOR MUTUAL LEARNING

We show additional results on the CKA between two GNN architectures—3-layer GAT and Graph-Sage—using the Citeseer dataset. Figure 7 (a) shows that the similarities between layers $1/2/3$ of the GAT architectures are $0.4/0.078/0.34$, while Figure 7 (b) shows that the similarities between the layers of the GAT and GraphSage architectures are $0.36/0.35/0.11$.

# G    ADDITIONAL EXPERIMENT

|  |  | Node Classification | Graph Classification |
|---|---|---|---|
| Models | Methods | Citeseer | Ogbg-molbbbp |
| | Ind | $75.79 \pm 0.76$ | $82.95 \pm 1.25$ |
| | GML | $76.29 \pm 0.64$ | $82.95 \pm 1.41$ |
| GCN-GAT-T | GML-Co | $76.27 \pm 0.59$ | $83.67 \pm 0.91$ |
| | GML-W | $76.21 \pm 0.31$ | $\mathbf{85.05 \pm 1.51}$ |
| | GML-C | $\mathbf{76.51 \pm 0.52}$ | $83.80 \pm 1.00$ |

Table 4: Additional results for the classification performance of mutual learning for node and graph classification. GCN-GAT-T denotes GML with GCN and GAT, using GAT as the target model.

| Models | Methods | Cora | Citeseer | PubMed |
|---|---|---|---|---|
| | Ind-MLP | $70.10 \pm 1.12$ | $67.88 \pm 0.53$ | $84.25 \pm 0.75$ |
| | KD without GML | $88.52 \pm 0.51$ | $78.92 \pm 0.23$ | $88.45 \pm 0.15$ |
| | KD + GML | $88.62 \pm 0.23$ | $78.36 \pm 0.13$ | $88.75 \pm 0.26$ |
| GraphSage-GCN-C | KD + GML-Co | $88.62 \pm 0.25$ | $79.12 \pm 0.25$ | $88.29 \pm 0.31$ |
| | KD + GML-W | $\mathbf{88.67 \pm 0.23}$ | $78.74 \pm 0.26$ | $88.46 \pm 0.22$ |
| | KD + GML-C | $88.40 \pm 0.29$ | $\mathbf{79.18 \pm 0.38}$ | $\mathbf{88.78 \pm 0.35}$ |
| | KD without GML | $85.28 \pm 0.14$ | $76.69 \pm 0.10$ | $88.75 \pm 0.25$ |
| | KD + GML | $83.87 \pm 0.13$ | $75.81 \pm 0.10$ | $88.46 \pm 0.22$ |
| GAT-GraphSage-T | KD + GML-Co | $84.33 \pm 0.31$ | $\mathbf{77.07 \pm 0.03}$ | $88.54 \pm 0.28$ |
| | KD + GML-W | $\mathbf{86.08 \pm 0.27}$ | $76.67 \pm 0.17$ | $\mathbf{88.96 \pm 0.23}$ |
| | KD + GML-C | $83.50 \pm 0.52$ | $76.61 \pm 0.50$ | $88.49 \pm 0.31$ |
| | KD without GML | $85.28 \pm 0.14$ | $76.69 \pm 0.10$ | $88.75 \pm 0.25$ |
| | KD + GML | $86.03 \pm 0.29$ | $\mathbf{78.48 \pm 0.29}$ | $88.38 \pm 0.23$ |
| GCN-GAT-T | KD + GML-Co | $85.25 \pm 0.22$ | $77.88 \pm 0.33$ | $88.27 \pm 0.35$ |
| | KD + GML-W | $85.63 \pm 0.17$ | $78.06 \pm 0.14$ | $\mathbf{88.75 \pm 0.14}$ |
| | KD + GML-C | $\mathbf{87.32 \pm 0.33}$ | $77.84 \pm 0.25$ | $88.45 \pm 0.36$ |

Table 5: Additional results for the performance of MLP with KD using the target model. We use models with different architectures for the mutual learning process.

We present additional findings of the graph mutual learning process in Table 4, where we cooperatively train GCN and GAT models, with GAT as the target model instead of GCN. We utilized the Citeseer dataset for node classification and the Ogbg-molbbbp dataset for graph classification. The results demonstrate the effectiveness of our enhancements in improving mutual learning performance. Furthermore, we explored switching the target model for the knowledge distillation (KD) process, employing models with diverse architectures. The results are detailed in Table 5. The experiments were conducted using the Cora, Citeseer, and PubMed datasets. Our findings illustrate that our enhancements enable switching the teacher model for KD while still achieving performance gains compared to baseline models without cooperative training.

# H    EXPERIMENT ON OGBN-ARXIV

We conducted additional experiments on the larger ogbn-arxiv dataset from the OGB benchmark. This dataset is more representative of real-world graph scenarios while remaining within our computational budget. In this experiment, we used the GAT model as the target and evaluated GML's performance across varying cohort sizes with different initializations. The results, presented in Table 6, indicate that GML performs effectively on larger datasets, maintaining its ability to improve accuracy and

generalization. However, as the number of participating models increases, training time and memory usage also grow. We propose that this computational overhead can be mitigated through parallelization if sufficient GPUs with adequate memory are available. These findings provide a foundation for future exploration into scaling online collaborative learning techniques with GNNs to even larger datasets.

| # Models | Mean Accuracy (%) | Mean Time (s) | Mean Memory (MB) |
|---|---|---|---|
| 1 | 53.93 ± 0.21 | 153.801 ± 55.931 | 270.47 ± 12.99 |
| 2 | 54.00 ± 0.16 | 389.386 ± 107.798 | 270.55 ± 12.97 |
| 3 | 54.03 ± 0.15 | 1660.993 ± 1184.938 | 270.69 ± 11.62 |
| 5 | 54.05 ± 0.09 | 2366.298 ± 608.315 | 270.85 ± 11.58 |

Table 6: Performance Metrics Across Different Numbers of Models

## I   EFFECT OF NOISY GRAPH STRUCTURES ON GML.

| Methods | No graph | Random | Barabási-Albert |
|---|---|---|---|
| GCN-Ind | 89.00 | 70.00 | 69.33 |
| GCN-GAT-C | N/A | 71.30 | 76.30 |

Table 7: Comparison of GML with single model GCN when trained with different graph structures on the Iris dataset.

To demonstrate that our method can reduce the effect of overfitting to graph structures, we conducted an experiment using the Iris dataset, which is not inherently a graph dataset. We trained a GCN model with the Iris features and then created both a random graph and a Barabási-Albert graph for this dataset. The Barabási-Albert graph, generated through preferential attachment, is a scale-free graph. The results, presented in Table 7, showed a performance drop when training on these graphs, highlighting how GNNs can overfit to graph structures even when they are unnecessary for the classification task. More details on this issue can be found in Bechler-Speicher et al. (2023). Additionally, to illustrate the importance of our mutual learning approach, we repeated the experiment with the generated graphs using mutual learning. The results indicate significant performance improvements: from 70.00% to 71.30% with the random graph and from 69.33% to 76.30% with the Barabási-Albert graph. These improvements demonstrate that mutual learning helps GNNs leverage their collective knowledge to reduce overfitting to specific graph structures.

## J   IMPACT OF ADAPTIVE LOGIT WEIGHTING UNIT AND THE CONFIDENCE PENALTY MECHANISM

We tested two cohorts: one with similar architectures in the cohort and another with randomly sampled architectures from the three models utilized in our work. We designed this to minimize potential human bias. In our experiments, we executed the experiments 10 times, calculating the mean accuracy and standard deviation for each cohort. As shown in Figure 2, our results demonstrate that the integration of both the adaptive logit weighting unit and the confidence penalty significantly enhances performance compared to GML.

## K   SANITY CHECK

To further validate our findings, we applied the Wilcoxon signed-rank test on experiments in Appendix J, where the null hypothesis ($\mathbf{H_0}$: our model does not yield significantly better results than GML) was tested. With a significance level of $p < 0.05$, we obtained a $p$-value of $0.00098$ for both the mixed and same architecture cohorts. This result allows us to confidently reject $\mathbf{H_0}$, confirming that the improvements observed are statistically significant and not due to random fluctuations. These results underscore the substantial impact of the adaptive logit weighting unit and the confidence penalty mechanism.

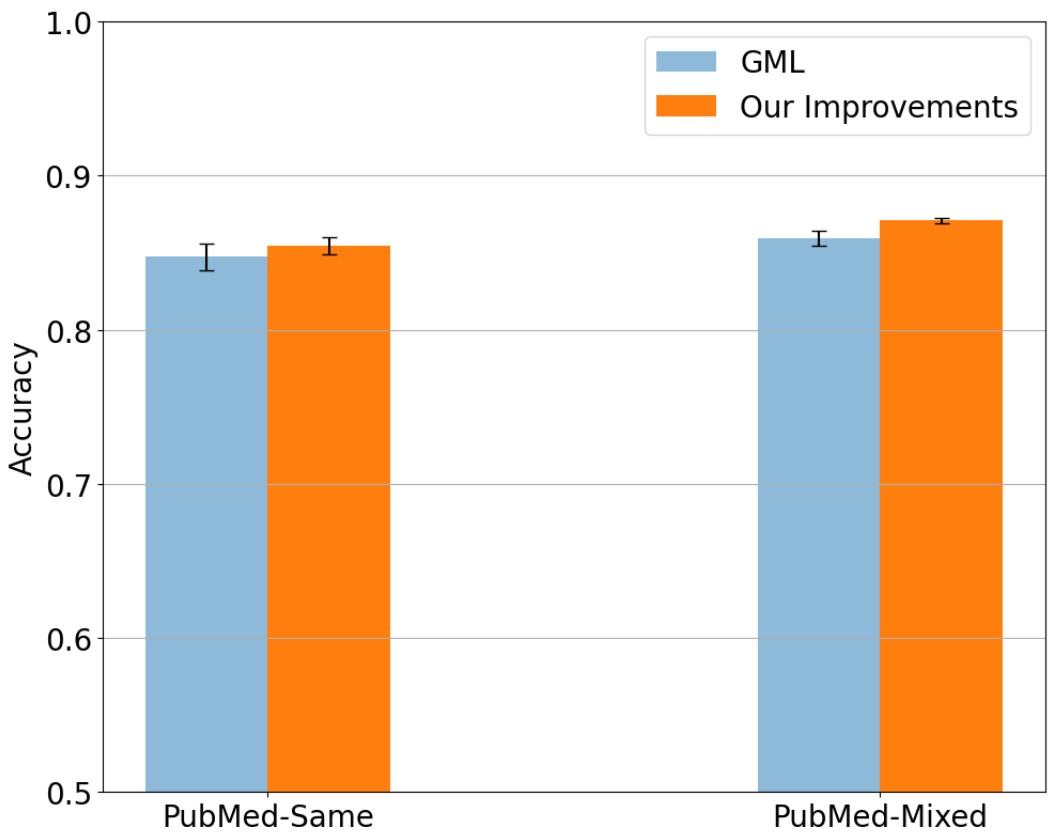

Figure 8: Comparison of our improvements over GML.

## L  IMPACT OF THE ENTROPY ENHANCEMENT COMPONENT UNDER NOISY CONDITIONS

To evaluate the impact of the entropy enhancement component under noisy conditions, we conducted additional experiments using GAT and GCN, with GCN as the target model. We introduced Laplace noise of varying magnitudes (0.1, 0.3, 0.5, and 0.9) to the node features and applied GML with the entropy enhancement component. The results, presented in Table 8, demonstrate the key insights.

**Improved robustness at mild to moderate noise levels**: Our findings shows that at noise levels 0.1, 0.3, and 0.5, GML with entropy enhancement component consistently outperformed individually trained GNN models. This improvement highlights the collaborative nature of GML, where models share complementary knowledge, effectively mitigating the impact of noise and preserving generalization ability, even under perturbations.

**Diminished effectiveness at high noise levels**: At higher noise magnitudes (e.g., 0.7 and 0.9), the noise become dominant, reducing the effectiveness of knowledge transfer and entropy enhancement. While GML still shows some robustness compared to individual models, the performance gap narrows as the noise levels increases.

## M  GRAPH-AWARE ADAPTIVE LOGIT WEIGHTING

Our current GML design achieves (i) Enhanced Expressiveness: By focusing on knowledge transfer among GNNs inherently suited for graph structures, our framework enables the ensemble to capture richer and more diverse graph representations. (ii) Adaptive Learning: We utilize techniques like adaptive logit weighting and entropy enhancement to optimize knowledge exchange among GNN

| Noise Level | GML-C (%) | GML (%) | Individual GCN (%) |
|---|---|---|---|
| No noise | $89.16 \pm 0.28$ | $88.77 \pm 0.57$ | $86.35 \pm 0.13$ |
| 0.1 | $82.86 \pm 0.55$ | $80.64 \pm 0.57$ | $79.88 \pm 0.29$ |
| 0.3 | $82.12 \pm 0.64$ | $80.76 \pm 0.76$ | $80.81 \pm 0.27$ |
| 0.5 | $81.40 \pm 0.48$ | $78.77 \pm 0.76$ | $81.87 \pm 0.44$ |
| 0.7 | $77.02 \pm 0.69$ | $77.32 \pm 0.90$ | $79.01 \pm 0.42$ |
| 0.9 | $82.41 \pm 0.33$ | $80.52 \pm 0.65$ | $83.03 \pm 1.00$ |

Table 8: GCN Results Across Different Setups and Noise Levels

peers. These components dynamically adjust learning strategies, particularly benefiting shallow GNNs that may lack the capacity of deeper models in other domains. Through collaboration, these models enhance their performance and generalization. However, GML can be extended to leverage the message-passing capability of GNNs seamlessly. To illustrate the potential for leveraging message-passing mechanisms more explicitly, we extended GML by incorporating a graph convolutional layer into the entropy computation process. Specifically, we refined the entropy values using neighborhood information via graph convolution. The negative entropy $H(p_j)$ is passed through a graph convolutional layer:

$$H_g(p_j) = \text{GCNConv}\big(H(p_j), \text{edge\_index}\big), \tag{18}$$

where $H_g(p_j) \in \mathbb{R}^{N \times 1}$ represent graph-convolved entropy, incorporating information from neighboring nodes and edge_index denote the adjacency list representing the graph structure. The importance score for class $c$ is then computed using the graph-convolved entropy:

$$\sigma_j^c = H_g(p_j)^\top \chi_j \phi_j. \tag{19}$$

The adaptive weight vector remains the same but now uses the revised importance scores. This refinement enables GML to utilize message-passing explicitly, enriching the knowledge exchange by considering graph topology during the computation of importance scores.

In our experiment experiment using the GCN-SAGE-S architecture on the Cora dataset, we observed improved performance with this extension. The results, presented in Table 9, demonstrate that integrating message-passing into GML further enhances its effectiveness for graph-specific tasks.

| Dataset | Best Accuracy Before (%) | Accuracy After (%) |
|---|---|---|
| Cora | $88.69 \pm 0.38$ | $89.48 \pm 0.42$ |
| Citeseer | $77.80 \pm 0.51$ | $78.78 \pm 0.32$ |
| PubMed | $90.19 \pm 0.30$ | $90.31 \pm 0.24$ |

Table 9: Comparison of Accuracy Before and After Graph-Aware Optimization for Different Datasets

# N  TRADE-OFFS BETWEEN MODEL DIVERSITY AND COMPUTATIONAL DEMANDS

As highlighted in Figure 5, we extended the GML framework to cohorts of up to 10 GNNs. These results demonstrate consistent improvements in accuracy and generalization as the number of participating models increases, showcasing the scalability of GML.

We further investigate the computational costs of GML compared to traditional unidirectional KD methods. While GML does not require a pre-trained teacher model, it does introduce additional computational overhead due to the collaborative training process. To quantify this, we conducted additional experiments on the Citeseer dataset, varying the number of peer models in GML and measuring training time, memory usage, and performance. To compare with unidirectional KD, we used the same GML framework but modified the knowledge transfer mechanism from bidirectional to unidirectional. The results, summarized in Table 10, reveal that while increasing the number of models improves accuracy and generalization, it also increases computational demands (time and

memory). However, these demands can be mitigated with parallelization. For instance, by aligning the number of participating GNNs with available GPUs, training time can be significantly reduced. This highlights GML's practicality even in resource-constrained settings, as it can leverage modern hardware to balance model diversity and computational efficiency.

| # Models | Mean Accuracy (%) | Mean Time (s) | Mean Memory (MB) |
|---|---|---|---|
| | GML | | |
| 1 | $64.86 \pm 3.28$ | $0.783 \pm 0.002$ | $117.998 \pm 0.031$ |
| 2 | $65.41 \pm 1.67$ | $1.930 \pm 0.075$ | $121.821 \pm 0.032$ |
| 3 | $70.06 \pm 0.91$ | $3.214 \pm 0.088$ | $127.683 \pm 0.032$ |
| 5 | $70.40 \pm 1.66$ | $7.433 \pm 0.114$ | $131.541 \pm 0.032$ |
| 9 | $71.67 \pm 0.29$ | $18.155 \pm 0.303$ | $164.438 \pm 0.032$ |
| 12 | $71.96 \pm 0.54$ | $32.733 \pm 0.604$ | $171.965 \pm 0.032$ |
| | Unidirectional KD | | |
| 1 | - | $0.500 \pm 0.054$ | $108.035 \pm 0.032$ |
| 2 | - | $1.401 \pm 0.040$ | $124.779 \pm 0.032$ |
| 3 | - | $2.407 \pm 0.005$ | $123.759 \pm 0.032$ |
| 5 | - | $4.628 \pm 0.007$ | $131.984 \pm 0.032$ |
| 9 | - | $8.006 \pm 0.006$ | $156.583 \pm 0.032$ |
| 12 | - | $12.006 \pm 0.013$ | $156.982 \pm 0.032$ |

Table 10: Performance Metrics Across Different Numbers of Models for GML and Unidirectional KD

## O    IMPACT OF MUTUAL LEARNING ON NODE CLASSIFICATION AND GRAPH CLASSIFICATION.

While our evaluation covers both node and graph classification tasks (as shown in Table 1), our analysis reveals subtle differences in GML's impact at these levels. For node classification, the maximum performance gain (+2.44%) slightly exceeds that for graph classification (+2.32%). This suggests that GML is particularly effective for localized tasks, such as node classification, where learning fine-grained, node-specific features and their immediate neighborhood structures is crucial. In contrast, graph classification, which depends on capturing holistic, global structural representations, also benefits from GML, albeit to a slightly lesser extent. This difference highlights a potential area for optimization to further enhance GML's capability to model and transfer global graph properties effectively. Additionally, we observed that the standard deviation for node classification (0.38) is significantly smaller than that for graph classification (1.67). This indicates that GML provides more stable and consistent performance improvements for node-level tasks, possibly due to the localized nature of mutual learning among peer models.

## P    LIMITATIONS AND FUTURE DIRECTIONS

This section highlights the framework's strengths while addressing key limitations and potential improvements.

- **Increased Computational Costs:** Training larger GML cohorts demands significantly higher time and memory resources, which can limit feasibility for extensive deployments.

- **Scalability Challenges:** Although performance improves with additional participating GNNs, the corresponding rise in computational demands poses difficulties for large-scale implementations.

- **Model Diversity Trade-Off:** Achieving optimal diversity within the cohort requires careful architectural selection, adding complexity to the design process.

- **Knowledge Transfer Limitations:** While GML consistently improves peer-to-peer GNN collaboration in graph classification, its effectiveness in transferring knowledge from GNNs

to MLPs for graph-level tasks remains suboptimal. This highlights the need for enhanced methods to better capture and transfer global structural properties, a focus of future work.

To address these challenges, potential improvements should include techniques that can address the challenge of leveraging knowledge gained through GML during knowledge transfer to MPLs, leveraging parallel processing with multiple GPUs, employing model compression techniques, and optimizing the trade-off between model diversity and computational efficiency.

