# OpenReview forum: "Enhancing Graph Neural Networks: A Mutual Learning Approach"
_ICLR.cc/2025/Conference — Submitted to ICLR 2025_

### Official Review · Reviewer_jHM7 · 2024-11-01

**Soundness:** 2
**Presentation:** 2
**Contribution:** 2
**Rating:** 5
**Confidence:** 5

**Summary:**

This paper introduces a novel approach called **Graph Mutual Learning (GML)**, which facilitates mutual learning among Graph Neural Networks (GNNs) to improve the performance of shallow GNNs without the need for a pre-trained teacher model. The GML framework involves collaborative learning among multiple GNNs, where models mutually teach each other throughout the training process. Key innovations include an adaptive logit weighting unit for effective knowledge exchange and an entropy enhancement technique to improve generalization. Extensive experiments demonstrate the effectiveness of GML on both node and graph classification tasks across various datasets. The authors also show that GML can further enhance knowledge distillation for downstream applications involving Multi-Layer Perceptrons (MLPs).

**Strengths:**

1. The GML framework introduces an adaptive logit weighting unit that prioritizes essential knowledge, complemented by entropy-based regularization to maintain effective uncertainty levels during training.
2. The elimination of the need for a pre-trained teacher model reduces dependency on large, pre-existing models, making the approach more accessible and flexible.

**Weaknesses:**

1. Although the evaluation covers both node and graph classification tasks, the paper lacks a detailed analysis of the specific impact of mutual learning on different graph learning levels. Does mutual learning benefit graph-level representation learning more than node-level representation learning? Understanding this aspect would highlight GML’s effectiveness for its target applications.
2. Mutual learning is a well-established concept in machine learning. However, when applied to graph neural networks, how does the mutual learning approach in this design differ from traditional mutual learning? The paper would benefit from more insights into whether the design is graph-specific or leverages message-passing mechanisms uniquely. Otherwise, it may appear as a straightforward application of mutual learning to new datasets.
3. The scalability of the proposed method remains unclear. For example, the node-level classification tasks use relatively small datasets such as Cora, Citeseer, and PubMed. Providing insights on the method’s scalability to larger datasets, such as ogbn-arxiv or ogbn-products, would be valuable. This suggestion also applies to the graph-level classification tasks.

**Questions:**

see weakness part

---

### Official Review · Reviewer_Z9xC · 2024-11-02

**Soundness:** 2
**Presentation:** 2
**Contribution:** 2
**Rating:** 5
**Confidence:** 3

**Summary:**

This paper introduces a new approach for enhancing the performance of GNNs through Graph Mutual Learning (GML). The method aims to have multiple GNNs collaboratively learn from each other, which is different from a traditional knowledge distillation paradigm where a teacher-student setup transfers knowledge from one model to the other. Key novelties of the paper include an adaptive logit weighting and an uncertainty enhancement mechanism that improves the quality of mutual learning, which prioritizes critical knowledge and promotes appropriate uncertainty in model outputs. Evaluations on node and graph classification datasets demonstrate that GML improves baseline GNN performance. Additionally, GML’s application to knowledge distillation for training lightweight MLPs suggests promising extensions of this framework for graph-less scenarios for fast inference without graph dependency.

**Strengths:**

- The mutual learning approach provides a new angle on GNNs knowledge sharing, expanding the conventional KD paradigm. This collective approach allows models to generalize more effectively without a pre-trained teacher.

- Experimental results on both node and graph classification tasks demonstrate the capacity of GML to improve performance. The MLP adaptation for KD highlights the versatility of the approach and the benefits of faster inference.

- The paper is clearly written and easy to follow.

**Weaknesses:**

- The experiments rely on standard datasets with relatively small sizes (e.g., Cora, Citeseer, OGB-bace and bbbp, etc), which may not fully capture the potential of GML on larger, real-world graph datasets. Including experiments on large-scale or complex datasets would strengthen the paper’s impact. For example, OGB provides much larger datasets for both node and graph classification.

- The extension from traditional teacher-student distillation to two peers collaborative learning is interesting, but also incremental. If the paper can discuss the generalized case with K GNNs more thoroughly, that would be better.

- A discussion of limitations can be beneficial.

**Questions:**

- How does the computational cost of GML compare with traditional KD methods in terms of training time and memory usage?

- As discussed in Appendix B, in extending the GML framework from 2 to K GNNs, how does the performance scale in terms of accuracy, generalization, and computational efficiency? Specifically, could you elaborate on any observed trade-offs between model diversity and computational demands as the number of peer models increases?

- The provided source code link seems to be broken.

---

### Official Review · Reviewer_dQwk · 2024-11-04

**Soundness:** 3
**Presentation:** 2
**Contribution:** 2
**Rating:** 3
**Confidence:** 2

**Summary:**

This paper introduces a collaborative learning framework, Graph Mutual Learning (GML), for enhancing shallow Graph Neural Networks (GNNs). Unlike traditional knowledge distillation from a teacher to a student model, this approach allows multiple GNN models to learn from each other in parallel. The proposed method includes adaptive logit weighting and uncertainty enhancement techniques to improve mutual learning and knowledge transfer. Experiments demonstrate the approach's effectiveness across several datasets in node and graph classification tasks.

**Strengths:**

1 - The concept of mutual learning among GNNs without a teacher model is innovative and potentially impactful for improving shallow GNN models.

2 - The adaptive logit weighting and entropy-based uncertainty enhancement components are well-motivated for improving model generalization and adaptability.

3 - The paper provides extensive experimental results on multiple node and graph classification datasets, demonstrating the method's effectiveness in various scenarios.

**Weaknesses:**

1 - The paper lacks a comparison with state-of-the-art GNN models and other existing knowledge distillation techniques, which limits the understanding of how well the proposed approach performs relative to current advancements in GNNs.

2 - The experiments primarily focus on shallow GNN models without including stronger GNN architectures as baselines. Including state-of-the-art GNNs would provide a more robust evaluation.

3 - The quality of the figures is subpar, making them hard to read and interpret. Critical insights and comparisons are challenging to comprehend due to the low readability of the visual components.

**Questions:**

Have you considered evaluating GML on deeper or more advanced GNN architectures to better understand the scalability and applicability of your approach?

How does the proposed GML framework perform compared to state-of-the-art knowledge distillation techniques specifically designed for GNNs?

Could you clarify how the entropy enhancement component impacts the overall performance, particularly in scenarios with noisy data?

---

### Official Review · Reviewer_p3m7 · 2024-11-05

**Soundness:** 2
**Presentation:** 3
**Contribution:** 2
**Rating:** 3
**Confidence:** 3

**Summary:**

The paper explores a collaborative learning framework, Graph Mutual Learning, to enhance Graph Neural Networks by employing mutual learning techniques. This method allows multiple GNNs to improve by sharing knowledge during training, using an adaptive logit weighting scheme and an uncertainty enhancement technique to optimize learning efficiency. The approach also includes knowledge distillation to transfer learned knowledge from GNNs to simpler MLP models for faster inference. Experimental results across various datasets demonstrate improved performance in GNN models and successful knowledge transfer to MLPs.

**Strengths:**

The paper addresses the significance of enhancing GNNs through collaborative learning, which is essential for tasks requiring high generalization in node and graph classification. The presentation is easy to follow.

**Weaknesses:**

1. The innovative aspects of the proposed methods are limited. The distinction between GML and similar approaches, such as those in [2], needs to be more clearly articulated to highlight the unique contributions.

2. Important baselines, specifically state-of-the-art knowledge distillation methods ([1,2,3]), are missing from Table 1. Including these would provide a more comprehensive evaluation.

3. The paper should clarify if the performance improvements in Tables 2 and 3 result from the enhancements in the teacher model or the distinct approach of distilling knowledge to MLPs compared to GNNs. Why should MLP performance be a key focus in this paper? The effectiveness of GML could potentially be demonstrated solely with GNNs, without involving GNN-to-MLP distillation.

4. Given that GML is designed to improve model performance, it's predictable that GML-enhanced GNNs would outperform ensembles of non-enhanced GNNs. Further analysis could explore whether more than two collaborative GNNs in GML would yield even greater improvements.

I would consider raising my score if the authors provide a promising rebuttal.

[1] Graph-less Neural Networks, Teach Old MLPs New Tricks via Distillation. ICLR 2022.
[2] Boosting Graph Neural Networks via Adaptive Knowledge Distillation. AAAI 2023.
[3] MuGSI: Distilling GNNs with Multi-Granularity Structural Information for Graph Classification. WWW 2024.

**Questions:**

1. What are the specific differences between the adaptive logit weighting in this paper and the adaptive temperature technique in [1]?

2. What is the performance of GML when extended to three or more GNNs? Would it improve further?

[1] Boosting Graph Neural Networks via Adaptive Knowledge Distillation. AAAI 2023.

---

### Meta-Review · Area_Chair_6m7T · 2024-12-21

**Metareview:**

The paper proposes Graph Mutual Learning to enhance Graph Neural Networks by employing mutual learning techniques. The problem is important and popular. Experimental results demonstrate the method's effectiveness in various scenarios. The paper presentation is good. There are many issues raised by reviewers, such as the incremental novelty, missing important baseline methods, insufficient experiments, and lack of detailed analysis of the mutual learning on graph learning. Reviewers are negative about this work.

**Additional Comments On Reviewer Discussion:**

No discussion is necessary as all give negative scores.

---

### Decision · Program_Chairs · 2025-01-22

Reject